# The TinyV3RSE Hardware-in-the-Loop Vision-Based Navigation Facility

**DOI:** 10.3390/s22239333

**Published:** 2022-11-30

**Authors:** Paolo Panicucci, Francesco Topputo

**Affiliations:** Department of Aerospace Science and Technology, Politecnico di Milano, Via Giuseppe La Masa, 34, 20156 Milano, Italy

**Keywords:** optical camera testing, vision-based navigation, hardware-in-the-loop simulations, optical test bench, verification and validation

## Abstract

The increase in number of interplanetary probes has emphasized the need for spacecraft autonomy to reduce overall mission costs and to enable riskier operations without ground support. The perception of the external environment is a critical task for autonomous probes, being fundamental for both motion planning and actuation. Perception is often achieved using navigation sensors which provide measurements of the external environment. For space exploration purposes, cameras are among the sensors that provide navigation information with few constraints at the spacecraft system level. Image processing and vision-based navigation algorithms are exploited to extract information about the external environment and the probe’s position within it from images. It is thus crucial to have the capability to generate realistic image datasets to design, validate, and test autonomous algorithms. This goal is achieved with high-fidelity rendering engines and with hardware-in-the-loop simulations. This work focuses on the latter by presenting a facility developed and used at the Deep-space Astrodynamics Research and Technology (DART) Laboratory at Politecnico di Milano. First, the facility design relationships are established to select hardware components. The critical design parameters of the camera, lens system, and screen are identified and analytical relationships are developed among these parameters. Second, the performances achievable with the chosen components are analytically and numerically studied in terms of geometrical accuracy and optical distortions. Third, the calibration procedures compensating for hardware misalignment and errors are defined. Their performances are evaluated in a laboratory experiment to display the calibration quality. Finally, the facility applicability is demonstrated by testing imageprocessing algorithms for space exploration scenarios.

## 1. Introduction

Recent discoveries of ocean worlds, scientific and economic interests in small bodies, and future human and robotic explorations are motivating numerous mission concepts aimed at Solar System exploration and exploitation [1]. The increasing number of interplanetary spacecraft is questioning the current approach to operate them. Currently, satellites are operated by ground-based engineering teams who communicate with the spacecraft throughout the Deep Space Network (DSN). The delayed communications and costs associated with human-in-the-loop interventions make this procedure unsuitable for supporting a massive and programmatic exploration of the Solar System.

Autonomous navigation would help overcome these limitations by providing real-time information to navigation filters by ensuring limited communications with the ground and low human-in-the-loop intervention. Amongst the different sensors used to perform autonomous navigation, cameras are usually preferred since they are light, compact, and require less power with respect to other advanced navigation sensors, such as Light Detection And Ranging (LiDAR) sensors. For these reasons, vision-based navigation (VBN), i.e., the combination of passive camera and image processing (IP) algorithms, provides compelling navigation performances with light and cost-effective hardware.

IP and VBN algorithms require images for design, performance assessment, and validation. It is thus crucial to generate realistic image datasets representative of the mission scenario and hardware characteristics. Unfortunately, in the context of interplanetary missions, the gathering of images during the algorithm design phase is a complex task. First, few missions have successfully flown in the past. This implies that available datasets are limited in number. Second, archived images are associated with spacecraft state estimates which are affected by orbit determination errors. Moreover, as outlined in Belgacem et al. [2], the use of spacecraft state estimates for image rendering leads to detectable errors in image generation. Third, the archived images are mission-specific. Archived images are associated with specific trajectories, mission constraints, and spacecraft characteristics. Thus, the available datasets could not be representative of the operational algorithm scenario. Therefore, image dataset generation must be carried out through other means. This need can be fulfilled by high-fidelity rendering engines or hardware-in-the-loop (HIL) simulations.

High-fidelity rendering engines are tools capable of generating images by simulating the physical behaviors of light, materials, and sensors. The simulation must not only include the propagation of light and its interaction with celestial bodies, but also the interaction of light with optical sensor components and its electronics [3]. These tools are usually convenient as several sensors can be implemented without expensive equipment at the cost of meticulous modeling efforts. Moreover, the true spacecraft trajectory is simulated at the software level, leading to a richer validation dataset. Several tools are available on the market to perform such simulations. Some, such as ESA’s PANGU [4] or Airbus Defence and Space’s SurRender [5], are specifically designed for space applications; however, they are not open source. Others, such as Blender [6] are open source, but lack physical realism as they are not designed for engineering applications. Despite their flexibility in simulating dynamic scenarios, a serious drawback of this approach is the procedure required to validate such a complex rendering framework.

On the contrary, HIL test benches rely on hardware testing in a laboratory environment to directly assess the performances of IP and VBN algorithms. The HIL facilities are generally more expensive, at least in terms of equipment, as the representation of the observed scene requires costly hardware such as robotic arms, optical components, or terrain analogs. Moreover, the use of hardware implies efforts in calibration that are not required when the sensor is simulated numerically. Despite these limitations, HIL facilities are usually preferred as hardware components represent mission operational errors. Because of this, TinyV3RSE, which stands for Tiny Versatile 3D Reality Simulation Environment, has been designed and developed to design, test, and validate IP and VBN algorithms with representative hardware errors. This work presents TinyV3RSE by focusing on its design, theoretical performances, functional workflow, geometrical calibration, and applications. This paper is an extension of previous work presented in Panicucci et al. [7] and Pugliatti et al. [8].

The rest of the paper is structured as follows. Section 2 presents the notation used in this work. Section 3 offers an overview of the HIL facilities for space application with a focus on optical facilities. Section 4 describes the facility design and analyzes its achievable performances. Subsequently, Section 5 illustrates the TinyV3RSE calibration procedure. Section 6 deals with calibration performances. In Section 7, two test cases are shown with applications for far-range and mid-range scenarios. Finally, Section 8 provides conclusions and future work for TinyV3RSE.

## 2. Notation

The following notation is used in this document:Two- and three-dimensional vectors are denoted, respectively, with lower- and upper-case bold text, such as r and R.Matrices are in plain text in brackets, such as $\left[A\right]$.Vector initialization is performed with parentheses, such as $b={\left(a^{T}\;a^{T}\right)}^{T}$.$A=\left\{a,\;a_{1},\;a_{2},\;a_{3}\right\}$ is the 3D reference frame centered in the 3D point *a* with axes $a_{1}$, $a_{2}$, and $a_{3}$. All the reference frames are right-handed and orthonormal.The rotation matrix from S to C is $\left[CS\right]$. All rotations have a passive function.The 3D vector from *s* to *c* is labeled $r_{c/s}$ and, when expressed in the S reference frame, is denoted ${}^{S}r_{c/s}$.$A=\left\{A,\;A_{1},\;A_{2}\right\}$ is the a 2D reference frame centered in the 2D point *A* with axes $A_{1}$ and $A_{2}$ which are orthonormal.The homography matrix from S to C is $\left[CS\right]$.The 2D vector from *S* to *C* is labeled $R_{C/S}$ and, when expressed in the S reference frame, is denoted ${}^{S}R_{C/S}$.The 3D vector r in homogeneous form is labeled ${}_{h}r$ and the 2D vector R in homogeneous form is labeled ${}_{h}R$.Projection of the 3D vector r on the 2D image is labeled R.The identity matrix of dimension *n* is labeled $\left[I_{n}\right]$. The zero n×m matrix is labeled $\left[0_{n\times m}\right]$ while the zero *n*-dimensional vector is labeled $0_{n\times 1}$.

## 3. Hardware-in-the-Loop Facilities for Space Navigation

### 3.1. Generalities on Hardware-in-the-Loop Facilities

In recent years, validation of complex algorithms and of advanced navigation sensors, i.e., cameras and LiDAR, has motivated the design and use of HIL facilities for space applications. Depending on their characteristics, HIL facilities for IP and VBN are categorized into two main groups: robotic and optical facilities.

Robotic facilities are composed of one or more robotic components, generally arms, where the sensor and the target mock-up are mounted on [9,10,11,12]. Light is simulated with a collimated lamp to emulate Sun illumination. The target mock-up is a scaled version of the space object under study whose coating must be consistent with the desired target albedo. This mock-up is illuminated by the lamp at the correct phase angle, while the relative observational geometry between the sensor and space object is simulated with the robotic components. In robotic facilities, the light interacts with the observed 3D mock-up. This is a crucial point for close-range VBN algorithms where local information, such as features, are used to navigate. Moreover, as the mock-up is continuous, the acquired image is not limited by the resolution of the rendered shape model or the screen stimulating the camera. Furthermore, using a mock-up drastically reduces the rendering time which is a time-demanding process for high-fidelity rendering. Nevertheless, several drawbacks are present. First, not all geometrical configurations are feasible due to the high volume of required hardware and its movement within the facility. Second, the development and use of the robotic facility is extremely costly, as it requires complex hardware and specialized operators. Third, this family of facilities is not suited to emulate deep-space scenarios, where space objects are not resolved.

An alternative solution is to use an optical facility. Optical facilities are test benches composed of a camera stimulated by a screen via a lens system. These facilities are more flexible than their robotic counterparts in reproducing different observational scenarios, from deep space to close proximity as the displayed image is generated at the software level. Moreover, the facility development costs are lower, since optical components are generally cheaper. Drawbacks are also present in this design. First, since a screen is observed, the camera does not capture a continuous world but a quantized one where pixels are the quanta. It is thus crucial to select the stimulating screen to minimize these effects and understand how to improve performance at the software level. Second, the lens system can induce distortions that warp the screen image observed from the camera. This distortion must be compensated at the software level and thoroughly analyzed to avoid systematic errors.

It is important to note that both facilities have the capacity, with their pros and cons, to reproduce mid-range and close-range observational geometry. Thus, the robotic and optical facilities are not exclusive in terms of applications, but can be used in a complementary and synergistic fashion. The optical facility can be exploited as a preparatory step for the robotic one in an incremental streamline within the verification and validation process. This step can ease the validation process in the robotic facility as the algorithm has already been tested on a representative scenario in the optical test bench.

### 3.2. Past Work on Optical Facilities

Historically, optical test benches were designed to characterize star trackers’ performances [13,14,15,16,17]. To the authors’ knowledge, the first optical facility detailed in the literature is outlined in Rufino and Moccia [18]. This work presents a collimated design composed of a cathode-ray tube display that stimulates a camera. This work focuses on the geometric and radiometric requirements to be fulfilled by the optical facility. Moreover, the attainable accuracy of a sensor mounted in the facility is assessed by considering uncertainties introduced by the HIL components. This aspect is further analyzed in Rufino and Moccia [13] by considering both static and dynamic simulations in the performance analysis. Recent work from Rufino et al. [16] presents the same facility with novel hardware, by analyzing the design with an LCD screen, attainable performances with the novel architecture, and results using COTS components.

The Jena-Optronik Optical Sky field sImulator (OSI) is presented in Samaan et al. [14] where this optical facility is used to stimulate a star tracker for the validation of attitude determination algorithms. The design employs a microdisplay to show star asterisms which are warped to compensate for lens distortion. Static and dynamic simulations are performed to validate the proposed attitude determination algorithm. The test bench is radiometrically and geometrically calibrated to correctly stimulate the camera. A similar solution is presented in Nardino et al. [15] to test a star tracker during spacecraft integration using an OLED microdisplay and a lens system to reduce the system volume.

This trend of miniaturization is also the main driver behind the optical test bench developed in Filipe et al. [17]. The facility is composed of an OLED smartphone display mounted on a rigid bar with a collimator. This work is mainly motivated by testing attitude determination algorithms in closed-loop simulations for CubeSats applications in deep space on an air-bearing system. Although the procedure is not detailed, the facility is geometrically calibrated and used for validation in MarCO and ASTERIA missions. This work also attempts to assess the radiometric calibration but with unsatisfactory results. Thus, the procedure is left for future work and the radiometric appearance of stars is performed by comparing similarity with deep space images.

In recent years, the importance of VBN for autonomous systems has motivated the application and development of optical facilities for VBN and IP on cameras [19,20,21].

The design presented in Roessler et al. [19] is inspired by the work of Rufino et al. [16] and presents applications not only for static and dynamic starry scenes but also in the context of spacecraft rendezvous. The system is composed of a LED monitor which stimulates a camera through a collimator. The geometrical calibration is achieved thanks to OpenCV library [22] that estimates the compensating homography by ignoring the collimator distortions.

An optical facility for inertial and far-range navigation is presented in Beierle et al. [20,21]. In this design, an OLED microdisplay is mounted on an optical breadboard to stimulate a camera. The test bench is geometrically and radiometrically calibrated to simulate stellar and non-stellar objects during far-range inertial navigation. The geometrical calibration compensates for distortion and linear errors achieving an accuracy of tens of arcseconds. The radiometric calibration is achieved through experimental analysis using an optical power meter by estimating the mapping between the screen luminance and the digital count value of the pixel. Results show the applicability of this design for far-range navigation. An extension of this work is presented in Beierle and D’Amico [23], where the optical facility is redesigned to obtain variable magnification, and thus variable camera field of views (FoVs), to tackle the problem of both far-range and close-proximity navigation.

## 4. Facility Design and Performances

### 4.1. Design Drivers

TinyV3RSE is composed of three main elements: a screen, a collimator, and a camera. The collimator is required to avoid working under the sensor hyperfocal distance when the focus is set near infinity [18]. Indeed, this ensures placing the screen image at infinity from the camera perspective. For a preliminary design, the thin-lens equation [24] can be used to model the behavior of light when passing through the collimator. Thus:(1)$$\frac{1}{f_{\mathrm{coll}}}=\frac{1}{d_{r}}+\frac{1}{d_{i}}$$
where $f_{\mathrm{coll}}$ is the collimating lens focal length, $d_{i}$ is the distance between the collimating lens and the image, and $d_{r}$ is the distance between the collimating lens and the object, i.e., the screen. Recall that $f_{\mathrm{coll}}$ is positive for converging lenses and negative for diverging lenses. Moreover, note that $d_{r}$ is positive when placed on the left side of the lens and negative otherwise. Finally, $d_{i}$ is positive when the image is generated on the right side of the lens, i.e., a real image is formed, and negative otherwise, i.e., a virtual image is formed. Figure 1 shows the geometrical configuration under study, which generates a virtual image.

Equation (Equation 1) can be rewritten to explicitly compute the image distance as:(2)$$d_{i}=\left(\frac{d_{r}}{d_{r}-f_{\mathrm{coll}}}\right)f_{\mathrm{coll}}$$

This shows that the object to be observed, i.e., the screen, must be placed at the collimator focal length distance if its image is to be put at infinity. This imposes the relative distance between the screen and the collimator but does not select which collimator focal length is required in the facility. This information can be gathered by imposing that the camera FoV perfectly fits the screen dimensions. This ensures that all information displayed on the screen is acquired from the camera without observing the screen support and frame. Hereunder, the vertical FoV is considered for simplicity since, in the current design, it is the narrower one.

Under the assumption of perfect components’ alignment, the problem can be studied as outlined in Figure 2. As the light rays are collimated, i.e., the angles are constant for all rays in the beam, it can be shown from simple geometrical consideration that:(3)$$\tan\left(\frac{\theta }{2}\right)=\frac{h_{s}}{2f_{\mathrm{coll}}}$$
where θ is the camera vertical full FoV and $h_{s}$ is the vertical screen size. Equation (Equation 3) establishes a relationship between the three components of TinyV3RSE, showing that their design choice is not arbitrary. In particular, it is worth noting that by changing one component, one of the other two must also be changed to verify Equation (Equation 3).

Note that because of the collimation, the distance between the camera and the collimating lens $d_{\mathrm{cam}}$ is not a design parameter that depends on the collimator focal length. This parameter is important for determining the diameter of the collimating lens [13]. To avoid the camera observing outside of the collimating lens, it must hold that [13]:(4)$$R_{\mathrm{coll}}\geq R_{\mathrm{cam}}+d_{\mathrm{cam}}\tan\left(\frac{\theta }{2}\right)$$
where $R_{\mathrm{coll}}$ is the radius of the collimating lens and $R_{\mathrm{cam}}$ is the radius of the camera lens objective entrance pupil. Moreover, to work in the paraxial area of the collimating lens, i.e., where the thin-lens equation approximation is valid, $d_{\mathrm{cam}}$ must be chosen to be as small as possible.

### 4.2. Components Selection

TinyV3RSE is composed of three main assemblies mounted on an optical breadboard and enclosed in a black box to limit light pollution from the external world:The camera assembly composed of the camera interface and its mechanical support that enables vertical translation, pitch, and yaw mechanical adjustments;The high-resolution screen assembly composed of the screen and its support, whose orientation is set to ensure that the screen and the optical plane of the camera are parallel;The collimator assembly composed of the collimating lens and its support. The collimator support comprises an optical holder that can rotate, change the height, and can be finely adjusted laterally and transversely.

These three assemblies are visible in the CAD model, shown in Figure 3 and from the top view of the TinyV3RSE facility in Figure 4. Each module is described in detail hereunder.

The camera used in this work is a Basler acA1300-22gm with a 12 mm fixed focal length lens objective. The key data-sheet characteristics of the camera are: a focal length of 12 mm, a resolution of 1280 pixels × 960 pixels, a pixel size of 3.75 μm × 3.75 μm, and a sensor size of 4.9 mm × 3.6 mm. The sensor mounted on the Basler camera body is a Sony ICX445 CCD sensor. The camera FoV is $22.6^{\circ }\times 17^{\circ }$. The camera is mounted on a dedicated assembly that is composed of three parts. The first one is a vertical translation stage ensuring vertical control of the assembly. The second one is a goniometer enabling pitch and roll. The third and last part is a custom adapter to interface the camera with the optical assembly.

The screen is represented by a Galaxy S7 smartphone with a resolution of 2560 pixels × 1440 pixels, a pixel size of 44.1 μm × 44.1 μm, and a screen size of 112.9 mm × 63.5 mm. As for the camera, the screen is mounted on a dedicated assembly composed of two parts. The first one is a translational stage that enables screen movement on a plane parallel to the optical breadboard. The second one is a screen-holding mechanism that enables re-orientation by changing four pins disposed close to the screen’s corners. The choice to use a commercial smartphone as a screen presents an advantage as it provides a low-cost solution with a compact high-resolution screen. The selected screen enables a reduction in distance between the screen and collimator, thus ultimately ensuring a compact facility. Therefore, the smartphone is a commercial, and hence low-cost, solution that also possesses interesting properties in terms of image contrast. The screen used is an OLED one, which does not suffer from screen bleeding phenomena typical of Liquid Crystal Display (LCD) screens and exhibits high contrast between inactive and active pixels. This is of particular interest in the rendering of the pitch-black background of a celestial scene.

With regard to the collimator, by considering the screen and camera characteristics and using Equation (Equation 3), the collimator should have a focal length of 211.7 mm. Because of this, the collimator has been chosen with a trade-off study among the available off-the-shelf components to maximize the observed portion of the screen while avoiding vignetting. Thus, the collimator used is a plano-convex lens with a focal length of 200 mm. The collimator is mounted on a dedicated assembly that is composed of a roto-translational stage and a post holder used to obtain vertical alignment between the collimator and the camera.

The screen, collimator, and camera assemblies are contained within a box that remains closed during image collection. This is done to ensure that proper illumination conditions are met and that no light artifacts are generated on the screen due to external conditions, i.e., reflections, external lighting, and shadows of personnel working next to the facility.

### 4.3. Functional Workflow

As stated in Section 1, HIL facilities aim at representing camera errors present in images without complex modeling of the camera system. Likewise, the hardware mounted in the facility should not be affected by systematic errors due to the components not being directly mounted on the spacecraft. Analogously, the facility requires meticulous calibration and compensation, as expounded in Section 5. Once this step is performed, the camera in the facility can acquire images that are corrected by systematic errors and can be used for algorithmic design, testing, and validation.

To do so, a functional workflow is put in place. The starting point is the simulated world, in which the physical and geometrical properties of the celestial bodies of interest are simulated in a virtual environment. The rendering of a scene can thus be seen as a sampling of this simulated synthetic environment through a simplified camera model. In this context, it is possible to identify three different images:The ideal mission image, which is the image that would be acquired by the modeled camera, labeled mission camera, in the virtual world.The screen image, which is acquired from the virtual world to be displayed on the screen. The camera rendering this image is labeled screen camera and is defined to match the screen resolution by keeping the FoV equal to the mission camera.The facility image, which is the image acquired in the facility by stimulating the facility camera.

In Figure 5, the functional workflow used to generate the aforementioned images is shown. On the left side of the sketch, the ideal rendering is depicted where the ideal mission image is generated. These images are the ones used for preliminary design and algorithm prototyping, and are also used to assess the algorithm’s feasibility. On the right side of the sketch, the HIL counterpart is depicted where the screen image is displayed on the screen and the facility image is acquired by the camera.

Note that the ideal mission image and the facility image are geometrically equivalent, yet photometrically different as HIL effects, such as defocus and errors, are present. The ideal mission image is generated via an ideal camera model, with no noise and perfect environmental conditions while the facility image encompasses noise and all phenomena typical of a sensor reading. The difference between these two images also represents the same domain gap between real and synthetic images that TinyV3RSE aims to reproduce for validation.

### 4.4. Achievable Performances with Selected Hardware

In this subsection, some considerations are gathered about the use of TinyV3RSE to represent a space scene with the selected hardware.

First, it is important to note that the screen has a size 112.9mm×63.5mm. By performing calculations, a coarse estimation gives that only 80mm×60mm of the screen is covered by the facility camera FoV, leading to about 100 screen pixels not observed by the camera. Thus, the image taken by the facility camera is just a portion of the image displayed on the screen.

Second, it is worth remarking that the screen has a finite resolution which implies that the displayed screen image is only a discrete representation of what the camera would observe during operation. It is possible to approximate a continuous world with a screen and a camera when the screen resolution is twice the camera resolution. Because of this, some facilities designed in the past seem to abide by this empirical sampling law for which each pixel of the sensor is to be stimulated at least by four pixels of the screen (or a 1:2 ratio if considered linear) [25,26]. However, at the time of the facility design, a higher resolution screen than the one considered has not been identified from existing commercial smartphones. The current setup has roughly a 1:1.5 ratio between sensor pixels and screen ones. If the loss of 100 pixels due to the reduced collimator focal length is considered, the ratio is reduced to 1:1.39. Note that this phenomenon did not seem to have produced a relevant disturbance in the performance of the tested algorithms.

Finally, as the pixels are finite in size, a feature angularly smaller than the pixel angular size as seen from the camera is approximated by a constant value within the pixel. Let $d_{\mathrm{px}}$ be the pixel size. It can be shown that the screen pixel angular dimension $\delta_{\mathrm{px}}$ as a function of the camera boresight offset $\alpha_{\mathrm{offset}}$ is given by:(5)$$\delta_{\mathrm{px}}=\tan^{-1}\left(\tan\alpha_{\mathrm{offset}}+\frac{d_{\mathrm{px}}}{2f_{\mathrm{coll}}}\right)-\tan^{-1}\left(\tan\alpha_{\mathrm{offset}}-\frac{d_{\mathrm{px}}}{2f_{\mathrm{coll}}}\right)$$

Under the assumption of a camera with narrow or medium FoV, Equation (Equation 5) can be approximated by the value at boresight:(6)$$\delta_{\mathrm{px}}≃2\tan^{-1}\left(\frac{d_{\mathrm{px}}}{2f_{\mathrm{coll}}}\right)≃\frac{d_{\mathrm{px}}}{f_{\mathrm{coll}}}$$

This is the angular size of the smallest features or objects to be displayed on the screen. Figure 6a shows the trend of $\delta_{\mathrm{px}}$ as a function of the offset angle $\alpha_{\mathrm{offset}}$ and underlines its low variability for the range of angles in the chosen camera FoV. For a pixel on the boresight, the value of $\delta_{\mathrm{px}}$ is 45.5 arcsec. This value is of crucial importance as it not only defines the angular size of the smallest object to be displayed, but also sets a limit on the tested IP and VBN algorithms’ precision without software preprocessing. Moreover, as shown by Rufino et al. [16], this value can be interpreted as the maximum uncertainty in the feature projection when a yaw-only or pitch-only rotation occurs. On the contrary, when a roll-only rotation occurs, the maximum uncertainty in the feature projection is given by [16]:(7)$$\epsilon_{\mathrm{px}}=2\tan^{-1}\left(\frac{d_{\mathrm{px}}}{2f_{\mathrm{coll}}\tan\alpha_{\mathrm{offset}}}\right)≃\frac{d_{\mathrm{px}}}{f_{\mathrm{coll}}\tan\alpha_{\mathrm{offset}}}$$
where $\epsilon_{\mathrm{px}}$ is the maximum angular uncertainty for a change in the apparent position of a feature for roll-only rotation. Figure 6b shows the change in $\epsilon_{\mathrm{px}}$ as a function of $\alpha_{\mathrm{offset}}$. This shows that the higher the offset angle, the lower the uncertainty. This implies that features and objects at the limits of the field of view are more informative in terms of roll, since they have lower uncertainty.

### 4.5. Distortion Analysis

In this subsection, a distortion analysis is performed to understand how the light rays are deformed when passing through the collimator. It is assumed in this analysis that the whole distortion is due to the collimator while the camera mounted in TinyV3RSE does not affect the light rays emitted from the screen pixels. This approximation is considered to be valid since cameras are generally designed to limit the distortion on the sensor. To perform this study, a Zemax^®^ model of the facility was developed as shown in Figure 7.

To understand the collimator distortion, several ray beams are defined by changing the position from which the rays are emitted. Each ray beam, which is composed of 1000 randomly oriented rays, is propagated through the system to impact the camera sensor plane. This defines the spot diagram shown in Figure 8.

Each subplot shows a spot diagram per ray beam of rays for the three main wavelengths, i.e., blue, green, and red, emitted from different positions on the screen. Note that the same colors are used in Figure 7 for different screen-emitting heights and in Figure 8 for different wavelengths. The ray beam generation position is reported on the top of each subplot. Note that the position at 30 mm in height and 40 mm in width is located at the camera FoV limits. This implies that the worst-case scenario for the ray distortion is presented. It can be noted that distortions affect how the screen pixels are seen from the camera. The rays starting from the screen center have a symmetrical distribution, but correct focus cannot be achieved for all three analyzed wavelengths. The spot radii increase by increasing the distance with respect to the sensor center which implies the presence of Petzeval field curvature. This is because the screen image, when passing through the lens, is curved, and thus cannot be focused on the camera sensor which is a plane surface. The spot diagram also shows coma distortion at the FoV borders which implies that the projected screen pixels appear to have a tail in the image.

To quantify these effects, the field curvature for the tangential and sagittal planes and the distortion percentage are reported in Figure 9a,b. The analysis is reported only for the horizontal FoV as it is the widest one. Note in Figure 9a that the focal point varies by considering ray beams emitted from different screens distanced from the center. This implies, as shown in Figure 9a, that a point stimulating the camera close to the sensor borders tends to have a larger defocus than the one on the optical axis. Moreover, the current design does not show any astigmatism since there is almost no difference between the tangential and sagittal planes.

Despite the mentioned distortion effects, their effects are not predominant in TinyV3RSE’s design as the distortion is less than 0.2% in the worst-case scenario, as shown in Figure 9b. The most remarkable effect is the coma which tends to shrink the image of screen pixels located at the FoV limits. This effect is limited as the Root Mean Square (RMS) radius of the spot diagram, which is about 10 μm, implying that the light emitted from a single screen pixel falls inside a circle of three-camera-pixels’ radius, i.e., 11.25 μm.

## 5. Geometrical Calibration and Compensation

Before using the facility, it is necessary to compensate for artifacts due to testing on a laboratory test bench rather than in orbit. When the light emitted by the display passes through the collimator, it is distorted by the lens. Thus, the image displayed on the screen is warped by the collimator, therefore changing the observed image. Moreover, the erroneous alignment and centering of components must be removed as they introduce biases in terms of rotation and position on VBN and IP algorithms estimates. Before entering the details of the procedure, it is important to define the following nomenclature:Calibration is defined as the process of estimating and quantifying the effect of the artifacts. This step is generally put in place by operators before facility use.Compensation is defined as the process of modifying the TinyV3RSE data to counteract the artifacts’ effect.

### 5.1. Geometrical Calibration

As multiple phenomena cause the HIL-induced errors, the calibration problem is solved sequentially. First, the facility camera is calibrated to find the equivalent pinhole camera model and its intrinsic matrix. Different algorithms can be used to perform this task depending on the application and the available calibrating pattern. Samaan et al. [27] proposes a calibration method from star images that could be used before mounting the camera in the facility. Since a dedicated test campaign to acquire images of the sky was not possible, the algorithm proposed in Zhang [28] is used to estimate the camera intrinsic matrix by ensuring that the calibration patterns are located beyond the sensor hyperfocal distance. Once the facility camera model is determined, the artifacts due to the TinyV3RSE components are estimated.

Therefore, two distortion models are estimated by projecting a series of calibration dot patterns on the screen and acquiring them through the camera. By knowing the calibration dot position on the screen images and the camera ones, the distortion models are estimated by least squares. Tang et al. [29] introduces various mathematical representations to model the optical distortion. Among these, polynomial distortion provides precise results that are compatible with arcseconds precision required for space applications [14,23].

Let $S=\left\{S,\;S_{1},\;S_{2}\right\}$ and $F=\left\{F,\;F_{1},\;F_{2}\right\}$ be the 2D reference frames associated with the screen and facility images, respectively. Let $S=\left\{s,\;s_{1},\;s_{2},\;s_{3}\right\}$ and $F=\left\{f,\;f_{1},\;f_{2},\;f_{3}\right\}$ be the 3D reference frames associated with the screen and facility cameras, respectively. Moreover, let $\left[K_{s}\right]$ and $\left[K_{f}\right]$ be the intrinsic matrices associated with the screen and facility cameras, respectively. Figure 10 shows an overview of the different reference frames.

Let $P_{i,\;j}$ and ${}_{h}P_{i,\;j}$ be the *i*th dot in the *j*th calibration pattern in non-homogeneous and homogeneous coordinates, respectively. More details regarding non-homogeneous and homogeneous coordinates can be found in Hartley and Zisserman [30]. The homogeneous coordinates of $P_{i,\;j}$ expressed in the S and F reference frame are labeled ${}_{h}^{S}P_{i,\;j}$ and ${}_{h}^{F}P_{i,\;j}$. If no hardware-induced artifacts were present, the line of sight (LoS) $l_{i,\;j}$ gathered from the screen and facility images would be equal:(8)$${}^{S}l_{i,\;j}={}^{F}l_{i,\;j}$$
where ${}^{S}l_{i,\;j}$ and ${}^{F}l_{i,\;j}$ are the representation of $l_{i,\;j}$ in the S and F reference frame, respectively. It is important to recall that S is a reference frame existing in the virtual world, while F is a reference frame in the real world. Thus, they are quantitatively equal but conceptually different. Equation (Equation 8) is rewritten by exploiting projective geometry to remove dependence on the camera intrinsic matrix [30]:(9)$${}^{S}\ell_{i,\;j}=\lambda_{s}{}^{S}l_{i,\;j}={\left[K_{s}\right]}^{-1}{}_{h}^{S}P_{i,\;j}={\left[K_{f}\right]}^{-1}{}_{h}^{F}P_{i,\;j}=\lambda_{f}{}^{F}l_{i,\;j}={}^{F}\ell_{i,\;j}$$
where $\lambda_{s}$ and $\lambda_{f}$ are proportional constants used to impose the third component to one, i.e., $s_{3}^{T}\;{}^{S}\ell_{i,\;j}=f_{3}^{T}\;{}^{F}\ell_{i,\;j}=1$.

Because of facility systematic errors, Equation (Equation 9) no longer holds. These errors can be captured by fitting polynomial distortion models between ${}^{S}\ell_{i,\;j}$ and ${}^{F}\ell_{i,\;j}$. Two models can be fitted: the forward distortion model FD which maps ${}^{S}\ell_{i,\;j}$ to ${}^{F}\ell_{i,\;j}$ and the inverse distortion model ID which maps ${}^{F}\ell_{i,\;j}$ to ${}^{S}\ell_{i,\;j}$:(10)$${}^{S}\ell_{i,\;j}=\mathrm{ID}\left({}^{F}\ell_{i,\;j}\right)$$
(11)$${}^{F}\ell_{i,\;j}=\mathrm{FD}\left({}^{S}\ell_{i,\;j}\right)$$

For the sake of completeness, Figure 10 shows the functional relationship at the base of the calibration procedure.

By defining ${}^{F}x_{i,\;j}=f_{1}^{T}\;{}^{F}\ell_{i,\;j}$, ${}^{F}y_{i,\;j}=f_{2}^{T}\;{}^{F}\ell_{i,\;j}$, ${}^{F}z_{i,\;j}=f_{3}^{T}\;{}^{S}\ell_{i,\;j}$, ${}^{S}x_{i,\;j}=s_{1}^{T}\;{}^{S}\ell_{i,\;j}$, ${}^{S}y_{i,\;j}=s_{2}^{T}\;{}^{S}\ell_{i,\;j}$, and ${}^{S}z_{i,\;j}=s_{3}^{T}\;{}^{S}\ell_{i,\;j}$, the two models can be expressed in components:(12)$${}^{F}x_{i,\;j}=\sum_{k=0}^{N}\sum_{l=0}^{k}a_{(k-l),\;l}\;{}^{S}x_{i,\;j}^{\left(k-l\right)}\;{}^{S}y_{i,\;j}^{l}$$
(13)$${}^{F}y_{i,\;j}=\sum_{k=0}^{N}\sum_{l=0}^{k}b_{(k-l),\;l}\;{}^{S}x_{i,\;j}^{\left(k-l\right)}\;{}^{S}y_{i,\;j}^{l}$$
(14)$${}^{S}x_{i,\;j}=\sum_{k=0}^{N}\sum_{l=0}^{k}c_{(k-l),\;l}\;{}^{F}x_{i,\;j}^{\left(k-l\right)}\;{}^{F}y_{i,\;j}^{l}$$
(15)$${}^{S}y_{i,\;j}=\sum_{k=0}^{N}\sum_{l=0}^{k}d_{(k-l),\;l}\;{}^{F}x_{i,\;j}^{\left(k-l\right)}\;{}^{F}y_{i,\;j}^{l}$$
(16)$${}^{S}z_{i,\;j}={}^{F}z_{i,\;j}=1$$
where *N* is the maximal degree of the distortion model.

The goal of the calibration procedure is to solve for the constant coefficients $a_{(k-l),\;l}$, $b_{(k-l),\;l}$, $c_{(k-l),\;l}$, and $d_{(k-l),\;l}$. As the two models are linear in the coefficients, the problem is solved via least squares by observing several calibration dot patterns equally distributed over the camera plane.

It is interesting to note that, for N=1, the two distortion models represent affine transformations between the points on the two images, which is consistent with the collimating set-up designed for TinyV3RSE. Once the calibration is performed, it is used for the compensation step where TinyV3RSE images are modified to obtain a consistent output with the desired geometrical configuration.

### 5.2. Upstream Compensation

In the upstream compensation, the images are rendered from the virtual world by the screen camera and are warped before being displayed on the screen. The camera then acquires the warped screen image to obtain the facility image. These latter images are, by construction, corrected by HIL-induced errors. The whole process is presented in Figure 11.

Let $S^{\prime }=\left\{S^{\prime },\;S_{1}^{\prime },\;S_{2}^{\prime }\right\}$ and $S^{\prime }=\left\{s^{\prime },\;s_{1}^{\prime },\;s_{2}^{\prime },\;s_{3}^{\prime }\right\}=S$ be the 2D and 3D reference frames associated with the warped screen image. Let P be a general point on the image plane. The vector ${}^{S^{\prime }}\ell$ is the warped LoS on the unit plane which is computed as:(17)$${}^{S^{\prime }}\ell =\mathrm{ID}\left({}^{S}\ell \right)=\mathrm{ID}\left({\left[K_{s}\right]}^{-1}{}_{h}^{S}P\right)$$

From this vector, it is possible to compute the point on the warped image to be displayed on the screen:(18)$${}_{h}^{S^{\prime }}P=\left[K_{s}\right]{}^{S^{\prime }}\ell$$

When this point is displayed on the screen and acquired by the camera in TinyV3RSE, it is distorted by HIL components. The calibration process estimates this distortion to be modeled with the forward distortion model. Thus:(19)$${}^{F}\ell =\mathrm{FD}\left({}^{S^{\prime }}\ell \right)$$

By combining Equations (Equation 17) and (Equation 19), it is possible to note that
(20)$${}^{F}\ell =\mathrm{FD}\left({}^{S^{\prime }}\ell \right)=\mathrm{FD}\left(\mathrm{ID}\left({}^{S^{\prime }}\ell \right)\right)\approx {}^{S}\ell$$
where the last approximation is justified by $\mathrm{ID}\left(\;\cdot \;\right)\approx \mathrm{FD}{\left(\;\cdot \;\right)}^{-1}$.

Equation (Equation 20) states that facility LoSes on the unit plane ${}^{F}\ell$ are equal, up to calibration error, to the screen LoSes on the unit plane ${}^{S}\ell$. Analogously, by warping the screen image before displaying it on the screen, it is possible to compensate for the HIL errors and reestablish the equivalence in Equation (Equation 9). This implies that the facility camera in TinyV3RSE is observing the same LoSes of the screen camera in the virtual world.

### 5.3. Downstream Compensation

In the downstream compensation, the images are rendered from the virtual world and directly displayed on the screen to stimulate the facility camera. The image acquired by the facility camera is affected by the HIL-induced effects and thus it must be modified to compensate for these errors. This undistortion process is then performed downstream to the acquisition by unwarping the facility image. This process is depicted in Figure 12.

Let $F^{\prime }=\left\{F^{\prime },\;F_{1}^{\prime },\;F_{2}^{\prime }\right\}$ and $F^{\prime }=\left\{f^{\prime },\;f_{1}^{\prime },\;f_{2}^{\prime },\;f_{3}^{\prime }\right\}=F$ be the 2D and 3D reference frames associated with the warped facility image. As stated in Section 5.2, TinyV3RSE distorts the LoSes on the screen as the forward distortion model. Thus, the LoSes on the unit plane ${}^{S}\ell$ are modified as:(21)$${}^{F^{\prime }}\ell =\mathrm{FD}\left({}^{S}\ell \right)=\mathrm{FD}\left({\left[K_{s}\right]}^{-1}{}_{h}^{S}P\right)$$
where ${}^{F^{\prime }}\ell$ are the LoSes in the unit plane on the warped facility image. The homogeneous point in the warped facility image ${}_{h}^{F^{\prime }}P$ is
(22)$${}_{h}^{F^{\prime }}P=\left[K_{f}\right]{}^{F^{\prime }}\ell$$

To compensate for the distortion given by TinyV3RSE, the inverse distortion model can be used. Thus:(23)$${}^{F}\ell =\mathrm{ID}\left({}^{F^{\prime }}\ell \right)=\mathrm{ID}\left(\mathrm{FD}\left({}^{S}\ell \right)\right)\approx {}^{S}\ell$$

This implies that, also for the downstream calibration, the facility camera is observing the same LoSes of the screen camera in the virtual world.

### 5.4. Discussion on Compensations

After having introduced the two compensations used for TinyV3RSE, a general discussion is provided hereunder.

It was shown that the facility camera and the screen camera observe the same LoSes in the virtual world. It is important to determine where the facility camera is located in the virtual world and how it is oriented within it. This is not a trivial result as the facility camera does not belong to the virtual world but to the real world. By starting from Equations (Equation 20) and (Equation 23), it is possible to remark that:(24)$${}_{h}^{F}P\approx \left[K_{f}\right]{\left[K_{s}\right]}^{-1}{}_{h}^{S}P=\left[FS\right]{}_{h}^{S}P$$
where $\left[FS\right]$ is the homography from S to F.

Let ${}_{h}^{V}p$ be a homogeneous 3D point in the virtual world expressed in a convenient 3D reference frame $V=\left\{v,\;v_{1},\;v_{2},\;v_{3}\right\}$. Its projection ${}_{h}^{S}P$ in the screen in homogeneous coordinates is given by [30]:(25)$${}_{h}^{S}P=\left[K_{s}\right]\left[\left[SV\right]\;{}^{S}r_{v/s}\right]{}_{h}^{V}p=\left[K_{s}\right]\left[SV\right]\left[\left[I_{3}\right]\;{}^{V}r_{v/s}\right]{}_{h}^{V}p$$
where $\left[SV\right]$ is the rotation matrix from V to S and $r_{v/s}$ is the translation from *s* to *v*. Without loss of generality, it can be imposed that V=S, thus:(26)$${}_{h}^{S}P=\left[K_{s}\right]\left[\left[I_{3}\right]\;0_{3\times 1}\right]{}_{h}^{S}p$$
where $0_{3\times 1}$ is the three-dimensional zero vector. Analogously, the projection of point p in the facility image placed in the virtual world is given by:(27)$${}_{h}^{F}P=\left[K_{f}\right]\left[FS\right]\left[\left[I_{3}\right]\;{}^{S}r_{s/f}\right]{}_{h}^{S}p$$
where $\left[FS\right]$ is the rotation matrix from S to F and $r_{s/f}$ is the translation from *s* to *f*. By backprojecting the 2D point ${}_{h}^{S}P$ in the 3D space, the ray passing for ${}_{h}^{S}P$ and the camera position ${}_{h}^{S}r_{s/s}={}_{h}0_{3\times 1}$ is computed in homogeneous coordinates [30]:(28)$${}_{h}^{S}p\left(\mu \right)={\left(\left[K_{s}\right]\left[\left[I_{3}\right]\;0_{3\times 1}\right]\right)}^{†}\;{}_{h}^{S}P+\mu \left(\begin{array}{c} 0_{3\times 1}\\ 1 \end{array}\right)=\left[\begin{array}{c} {\left[K_{s}\right]}^{-1}\\ 0_{3\times 1}^{T} \end{array}\right]{}_{h}^{S}P+\mu \;{}_{h}0_{3\times 1}$$
where ${\left(\;\cdot \;\right)}^{†}$ is the Moore–Penrose inverse operation and μ is a parameter to sweep the ray. Note that ${}_{h}0_{3\times 1}$ is the origin in homogeneous coordinate, i.e., ${}_{h}0_{3\times 1}={\left(0_{3\times 1},\;1\right)}^{T}$.

Then, the backprojected ray can be projected into the facility camera as a function of μ:(29)$$\begin{array}{cc} {}_{h}^{F}P\left(\mu \right) & =\left[K_{f}\right]\left[FS\right]\left[\left[I_{3}\right]\;{}^{S}r_{s/f}\right]\left(\left[\begin{array}{c} {\left[K_{s}\right]}^{-1}\\ 0_{3\times 1}^{T} \end{array}\right]{}_{h}^{S}P+\mu \;{}_{h}0_{3\times 1}\right)=\\ \; & =\left[K_{f}\right]\left[FS\right]\left[\begin{array}{c} {\left[K_{s}\right]}^{-1}\\ 0_{3\times 1}^{T} \end{array}\right]{}_{h}^{S}P+\mu \;\left[K_{f}\right]\left[FS\right]\left[\left[I_{3}\right]\;{}^{S}r_{s/f}\right]\left(\begin{array}{c} 0_{3\times 1}\\ 1 \end{array}\right)=\\ \; & =\left[K_{f}\right]\left[FS\right]{\left[K_{s}\right]}^{-1}\;{}_{h}^{S}P+\mu \left[K_{f}\right]\left[FS\right]{}^{S}r_{s/f} \end{array}$$

To demonstrate that Equation (Equation 29) holds for all μ, i.e., each backprojection from the screen camera has a unique reprojection in the facility camera, it is necessary that $r_{s/f}=0_{3\times 1}$. Thus, the two cameras are located at the same point in the virtual world. Moreover, to obtain the correct homography between the two images as reported in Equation (Equation 24), it is also necessary that $\left[FS\right]=\left[I_{3}\right]$, i.e., the two cameras have the same orientation in the virtual world. It is possible to conclude that the two cameras have the same poses in the virtual world if Equations (Equation 20) and (Equation 23) hold.

Moreover, it is worth discussing the reasons behind developing two compensation methods. The upstream compensation implies image warping before image acquisition without modifying the facility camera image. The image is then directly transferred to the IP and VBN chain without any post-processing. Unfortunately, to apply the upstream calibration, this procedure must be implemented at the rendering level to ensure the correct display of the scene on the screen. This means that the virtual world must be warped before being projected on the rendering camera model. This process is not straightforward since the rendering engine must have this functionality. Custom software can be designed to embed this functionality, but this could limit the applications of TinyV3RSE to specific rendering scenarios. On the contrary, the downstream calibration warps the image after image acquisition. This implies that the compensation step introduced between the camera acquisition and the VBN pipeline could affect the verification and validation procedure. Nevertheless, this compensation enables the use of open-source software for rendering and performing the preliminary assessment of the algorithms’ performances without the need for developing a new rendering engine from scratch.

Finally, it is possible to use TinyV3RSE in two different ways. On the one hand, the VBN system—integrated in terms of hardware and software—can be verified and validated in TinyV3RSE. This implies the mounting of a camera engineering model in the facility, acquisition of the images, and execution of the algorithmic chain on a dedicated processor. On the other hand, TinyV3RSE can also be used to fast-prototype algorithms even with camera characteristics different to the one mounted in the facility. This is possible by modifying the acquired image after compensation to simulate other camera models. Let $M=\left\{m,\;m_{1},\;m_{2},\;m_{3}\right\}$ and $M=\left\{M,\;M_{1},\;M_{2}\right\}$ be, respectively, the 2D and 3D reference frames associated with the mission camera not physically present in the facility. Moreover, let $\left[K_{m}\right]$ be the intrinsic matrix associated with the mission camera. Note that this camera exists in the virtual world and that M=S, i.e., the two cameras have the same pose in the virtual world, as the final goal is to generate images with HIL noise by projecting images on the screen. By using Equation (Equation 29), the homography $\left[MS\right]$ from S to M is
(30)$$\left[MS\right]=\left[K_{m}\right]{\left[K_{s}\right]}^{-1}$$

Thus, after compensation, it is possible to use Equation (Equation 24) to map the image taken by the facility camera to the one taken by the mission camera:(31)$${}_{h}^{M}P\approx \left[MS\right]\left[SF\right]{}_{h}^{F}P$$
where $\left[SF\right]={\left[FS\right]}^{-1}$. Note that the facility image includes HIL effects which are also mapped to the mission camera image. Thus, mission camera images obtained with Equation (Equation 31) include the the camera errors. This can help to identify issues in the algorithmic pipeline at a preliminary stage of the design. Images obtained in this manner are presented in Figure 13a,b showing the difference between the perfectly rendered image in the virtual world and the image taken in the real world with HIL effects and then mapped to the mission camera.

## 6. Calibration and Compensation Performance

In this section, the performances of the geometrical calibration and compensation are analyzed. To do so, two performance indices are defined. The first one is the error in pixel location between the true position of the calibration pattern points and their observed counterparts. This index is computed to understand the statistical distribution in the image of the calibration and compensation errors. The error $\varepsilon_{i,\;j}$ for the *i*th dot on the *j*th calibration pattern is expressed as:(32)$$\varepsilon_{i,\;j}={}^{F}P_{i,\;j}-{}^{F}{\hat{P}}_{i,\;j}$$
where ${}^{F}P_{i,\;j}$ is the *i*th dot position in the facility image on the *j*th calibration pattern and ${}^{F}{\hat{P}}_{i,\;j}$ is the known pixel position for the *i*th dot on the *j*th calibration pattern illuminated on the screen and mapped from the screen image to the facility image. This latter quantity can be computed in homogeneous coordinates by using Equation (Equation 24) and by knowing the position of the illuminated pixel on the screen.

The second performance index is a global performance index which provides information on the errors distribution norm in angular dimension. Let $\theta_{\mathrm{IFoV}}^{f}$ and $\theta_{\mathrm{IFoV}}^{s}$ be the facility and screen Instantaneous Field of View (IFoV) which are the angular dimensions of a single pixel of the screen and facility camera, respectively. The IFoV error $\varepsilon_{{\mathrm{IFoV}}_{i,\;j}}$ for the *i*th dot on the *j*th calibration pattern is defined as
(33)$$\varepsilon_{{\mathrm{IFoV}}_{i,\;j}}=\left|\left|\varepsilon_{i,\;j}\right|\right|\theta_{\mathrm{IFoV}}^{f}\approx {}^{S}l_{i,\;j}^{T}{}^{F}l_{i,\;j}$$
where the last approximation holds because the iFoV is the ratio between the camera focal length and the camera pixel size. The IFoV error is a metric of the angular error between the observed LoSes and the expected ones. Note that $\varepsilon_{{\mathrm{IFoV}}_{i,\;j}}$ and $\varepsilon_{i,\;j}$ are scalars, the angular error for all the dots in all calibration patterns are stored in vectors labeled $\varepsilon_{\mathrm{IFoV}}$ and ε, respectively.

### 6.1. Calibration

In this section, the calibration performances are depicted on a numerical example. First, the facility is calibrated. Next, TinyV3RSE is calibrated by projecting 50 randomly oriented and randomly positioned calibration patterns composed of pixel-wise dots. The number of patterns is chosen to cover the whole facility camera image plane with calibrating dots. This ensures that distortions of the collimator are well estimated. Finally, the performance indices are computed for each calibration pattern that is not outside the facility image borders.

The Probability Density Functions (PDFs) and Cumulative Density Functions (CDFs) of the angular error $\varepsilon_{\mathrm{IFoV}}$ are shown in Figure 14a,b for the direct and inverse distortion models, respectively. Indeed, as shown in the CDFs, 99% of the points are under 20 arcsec angular error, which is less than half of the screen IFoV and one-third of the facility camera IFoV. This implies that the distortion models are correctly estimated since the calibrations dots are warped on their screen counterparts correctly. This can be also observed in Figure 15a,b where the two-dimensional PDFs are shown for the pixel error ε against the screen pixel size as seen from the facility camera and against the facility camera pixel. Errors are well below the facility camera and screen pixel size which implies a good fitting of the HIL effect. Moreover, to confirm these results, the first two stochastic moments of the pixel error, i.e., the mean and the covariance, are reported in Table 1. The error is close to zero with very low covariance matrix, which means that no outliers are present in the estimation and that the distortions models fit the calibration data well.

### 6.2. Upstream and Downstream Compensation

Once the calibration estimates the direct and inverse distortion models, the upstream and downstream compensation performances can be evaluated. To do so, the proposed compensations are tested on 100 calibration patterns which are randomly oriented, randomly positioned, and never seen from the facility camera. This set of calibration patterns is used to assess the performances of the calibration on a test set which is independent with respect to the calibration set. Before analyzing the compensation results, it is important to recall that all calibration dots within a pixel are equivalent as the screen cannot display a point under its resolution. Because of this, the compensation can be accurate up to the screen pixel angular size and the expected distribution of the errors, both in pixel and in angles, and is a uniform bivariate distribution centered in zero and whose area is defined by the screen pixel or solid angle, respectively.

The PDFs and CDFs of the angular error $\varepsilon_{\mathrm{IFoV}}$ are shown in Figure 16a,b for the upstream and downstream compensations, respectively. As shown in the CDFs, the angular errors are below the screen IFoV more than 99% of the time for both compensation methods. This implies that the points are warped correctly in both cases to observe similar LoSes in the virtual world and in the facility. This is true both when the image is warped before being displayed on the screen and when the image is distorted after being acquired by the camera. A deeper analysis of the results shows that the dots with error higher than the screen IFoV are generally located at the image border where the collimator distortion is higher. Moreover, as the compensation pattern dots are different from those used during calibration, the experimental results show that the distortion models generalize well the HIL effects to a test set that had previously never been observed by the facility camera. Indeed, the distortion models are computed by overfitting the calibration pattern dots, but they correctly capture the HIL errors with previously unprocessed data, i.e., compensation pattern dots.

The same consideration can be gathered by observing the two-dimensional PDF for the pixel error ε, shown in Figure 17a,b for the upstream and the downstream compensation, respectively. Errors are concentrated within the pixel error with a distribution close to a uniform bivariate distribution whose extension is the screen pixel size. This conclusion is also consistent with the two first moments of the PDF for both upstream and downstream compensations. The mean and covariance, shown in Table 2, are close to the expected uniform bivariate distribution, which is reported in the same table for the sake of completeness. The similarity between the two distributions has also been investigated by using the two-sample Kolmogorov–Smirnov test. The test concluded that the null hypothesis is rejected at 0.5% significance level. This confirms the expected theoretical results on an experimental example.

For these analyses, it is concluded that the facility camera extracted the same LoSes observed by the screen camera in the virtual world with a statistical distribution consistent with the hardware present in TinyV3RSE.

## 7. Applications

In this section, two IP algorithms are tested in TinyV3RSE using both upstream and downstream calibrations. First, the attitude determination algorithm employed within the EXTREMA project [32] is tested on starry sky images. Next, the IP pipeline developed for the ESA Milani’s mission [33] is tested on TinyV3RSE-acquired images to understand its sensibility to hardware noise.

### 7.1. Image Processing for Attitude Determination

The first presented application relates to the attitude determination algorithm developed within the EXTREMA project [32]. EXTREMA aims at enabling self-driving interplanetary CubeSats in deep-space cruise. To navigate throughout the Solar System, the probe can exploit the known position of planets at a given epoch and triangulate the spacecraft position when these planets are recognized in on-board camera images [34]. The IP pipeline which provides measurements to the navigation filter is divided in several steps [35]:Centroid extraction which provides the center of brightness of bright spots in the image.Star asterism identification which performs matching between the bright spots and the on-board star catalog.Attitude determination which solves the Wabha’s problem [36] by finding the probe orientation given a series of LoSes in the spacecraft-fixed reference frame and the inertial reference frame.Planet identification which performs matching between the inertial positions of the planets and their counterparts observed in the image.

Given this pipeline, a crucial step to provide accurate and precise information to the navigation filter is the on-board attitude determination algorithm. Within the context of EXTREMA, Andreis et al. [35] develops an IP pipeline for attitude determination based on the k-vector method [37] for star asterism identification and RANdom SAmple Consensus (RANSAC) algorithm [30] for the robust attitude determination procedure. This procedure is preliminary to ensure correct planet identification to star asterism mismatch as the attitude is directly used in the spacecraft-fixed reference frame calculations. Thus, errors in attitude determination can induce biases in the IP measurements given to the filter, and thus on the navigation. Because of this reason, it is interesting to test the proposed algorithm on hardware to assess the robustness to camera errors.

To do so, a batch of 1000 images is generated by randomly selecting the spacecraft attitude in a custom-designed deep-space rendering engine, which is an extension of the simulation environment proposed in Bella et al. [38]. The simulator can render images at the screen resolution by warping the LoSes of the celestial objects before image generation to include the upstream compensation directly at the rendering level. Radiometric calibration of TinyV3RSE has not been rigorously performed in the current study and is left for future development of the facility. The current pipeline is based on consideration of consistent appearance of the observed image in the facility camera. This approach is consistent to validate the IP and VBN pipeline from a observational geometry perspective. Since the camera is a radiometric sensor, it is important to calibrate the screen to receive the correct irradiance at the camera pupil. This can be achieved by knowing the mapping from pixel luminance to digital count [13] or by finding the mapping empirically [21,23]. In fact, the approach adopted in this paper, following Filipe et al. [17], assumes that this mapping is linear, which is inconsistent with experimental set-ups (see Figure 6 in Beierle et al. [21] and Figure 5 in Rufino and Moccia [13]). As these steps are left for future work, to associate a digital count to each star in the image, the procedure presented in Filipe et al. [17] is used. Images are subsequently displayed on the screen statically and are acquired by the camera to obtain the validation dataset. The images are then given as input to the IP algorithm to provide an estimation of the camera orientation and a series of identified stars in the image. For the sake of completeness, an example of an image taken by the facility camera is provided in Figure 18 in reversed colors. A subset of the observed stars are correctly identified in the image by matching the on-board catalog. The identified stars are shown in Figure 18 in blue boxes with the corresponding identifiers.

The Wabha’s problem is then solved by rejecting possible outliers in the stars identification procedure thanks to RANSAC. This provides per image the Direction Cosine Matrix (DCM) $\left[N\hat{B}\right]$ which is the DCM from the estimated spacecraft reference frame $\hat{B}=\left\{\hat{b},\;{\hat{b}}_{1},\;{\hat{b}}_{2},\;{\hat{b}}_{3}\right\}$ and the inertial reference frame $N=\left\{n,\;n_{1},\;n_{2},\;n_{3}\right\}$. By labeling [NB] the DCM from the true spacecraft reference frame $B=\left\{b,\;b_{1},\;b_{2},\;b_{3}\right\}$ and the inertial reference frame $N=\left\{n,\;n_{1},\;n_{2},\;n_{3}\right\}$, the DCM from $\hat{B}$ to B is:(34)$$\left[B\hat{B}\right]={\left[NB\right]}^{T}\;\left[N\hat{B}\right]$$
which is the error DCM between the estimated and true reference frame. The attitude is considered to converge to a feasible solution when the attitude error is below 200 arcsec. By considering this, the attitude determination has a success rate of 93.7% on the current dataset. Moreover, to represent the attitude error $\left[B\hat{B}\right]$ for the feasible cases, the roll–pitch–yaw Euler angles are shown in Figure 19. Results are consistent with the selected hardware, and the errors are bounded by the screen iFoV and the camera iFoV.

### 7.2. Image Processing for Small Body Navigation

The second application gives a glance at the validation and verification of the Milani’s IP and is reported to show the applicability of the downstream compensation procedure. This procedure, as explained in Section 5.4, is preferred when it is not possible to warp the LoSes during the rendering process. In this scenario, the images are generated using Blender as the core rendering engine wrapped in a procedural image generator for small-body application [39]. This study was first briefly reported on in Panicucci et al. [7] and Piccolo et al. [31] in more detail.

Milani [33] is a 6U CubeSat, part of the ESA’s Hera mission [40]. The Hera mission aims at studying and characterizing the Didymos binary system. The Hera spacecraft will release two CubeSats, Milani and Juventas, when in proximity of the binary system to perform scientific and technological demonstrations. Milani embeds semi-autonomous navigation functionalities, based on innovative IP and VBN, to enhance a traditional Guidance, Control and Navigation (GNC) approach. The mission profile foresees a Far-Range Phase (FRP) and Close-Range Phase (CRP) both composed of several flyby arcs, as shown in Figure 20a,b.

The semi-autonomous navigation functionalities rely on optical measurements extracted by the IP pipeline. The IP pipeline works as follows:The raw image is analyzed to identify the primary and the secondary. From this step, a set of geometrical characteristics of the corresponding pixel blobs, such as the centroid and the best-ellipse-fitting parameters, are computed.The so-called WCOB (Weighted Center Of Brightness) is applied to determine a data-driven weighted correction on the Center of Figure (CoF) to approximate the primary Center of Brightness (CoB). During this phase, the phase angle (PA) is also computed with data-driven functions.Finally, the range to the primary is estimated from an apparent diameter formula.

From the aforementioned IP algorithm, it is possible to gather the CoF as a data-driven correction of the CoB, a data-driven-derived PA, and the range. A more detailed discussion of the Milani’s IP can be found in Pugliatti et al. [41] and Pugliatti et al. [42].

As the coefficients of the data-driven functions are estimated from an image dataset rendered in a virtual world, it is crucial to understand how the WCOB performs with realistic hardware errors. Thus, images from FRP and CRP are rendered at the screen resolution and displayed on the screen to stimulate the camera mounted in TinyV3RSE. This results in an image dataset which include camera errors and calibration errors. For the sake of clarity, a comparison between ideal and facility-acquired images are reported in Figure 21a,b for the FRP and in Figure 22a,b for the CRP.

Next, the WCOB algorithm gathers IP estimates from both ideal mission images and the facility images to compare them against the true values of the IP estimates. To evaluate the WCOB algorithm performance, several indices are used. First, the error on the CoF is computed both in the x- and y-image components and in the norm. The latter is obtained as:(35)$$\varepsilon_{n}=\sqrt{\varepsilon_{x}^{2}+\varepsilon_{y}^{2}}$$
where $\varepsilon_{x}$ and $\varepsilon_{y}$ are the errors in the x- and y-components in the image plane. Moreover, the errors in the phase angle and range are labeled, respectively, as $\varepsilon_{\mathrm{PA}}$ and $\varepsilon_{\mathrm{Range}}$, and are computed as well. The difference between the ideal-image error and the facility-image error is labeled by placing a Δ before the corresponding variable (e.g., $\Delta \varepsilon_{x}$).

The obtained results are shown in Figure 23 and Figure 24 for the FRP and CRP, respectively. The CoF error is mostly below 20 pixels for both cases which is consistent with the 1-σ measurement noise given to the navigation filter [43]. The CoF error is higher for the CRP as the spacecraft is closer to the binary system which implies that the primary appears larger in the image. Moreover, the differences between ideal-image and facility-image estimates, i.e., $\Delta \varepsilon_{x}$, $\Delta \varepsilon_{y}$, and $\Delta \varepsilon_{n}$, are higher in the CRP. This is due to the fact that the algorithm is affected more by discrepancies in pixel intensity, mainly due to noises, when the primary appears larger (see Figure 21 and Figure 22). Furthermore, this difference in pixel content causes the PA and range to be slightly biased with respect to their ideal-image counterparts. Nevertheless, this is not crucial for Milani as the CoF is the only IP estimate directly used in the navigation filter.

## 8. Conclusions and Future Work

This work presents the hardware-in-the-loop optical facility developed at DART Lab to perform verification and validation of IP and VBN algorithms. First, the rationale behind the design is detailed and main design drivers are gathered with simple and analytical relationships leading to to the selection of the TinyV3RSE’s components. The selected components also define the achievable performances of the facility which are analyzed both in terms of geometrical accuracy and of optical distortions. Subsequently, this work presents two approaches, i.e., the upstream and downstream compensations, to limit the effects induced by errors in centering, component alignment, and optical distortion. Both compensations are studied from a theoretical perspective demonstrating that the image acquired is the same image as captured by the facility camera placed in the desired position and orientation in the virtual world. Moreover, experimental results are shown to assess the quality of both compensations. The results demonstrate agreement with the performed theoretical analysis and underline the possibility of observing a space object within the facility camera FoV as it would be observed from the same camera but in orbit. Achievable precision is bounded by the screen pixel size, which is the smallest quantum displayable on the screen. Finally, to show the applicability of TinyV3RSE for validation of space exploration scenarios, two scenarios are tested in TinyV3RSE: a deep-space attitude determination test case and a mid-range small-body test case.

## Figures and Tables

**Figure 1 sensors-22-09333-f001:**
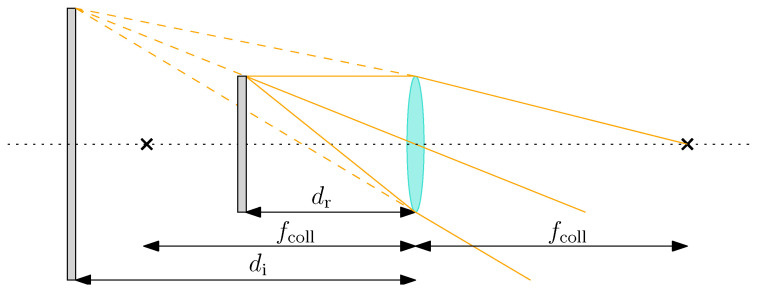
Geometrical configuration for the lens equation of the collimator in TinyV3RSE.

**Figure 2 sensors-22-09333-f002:**
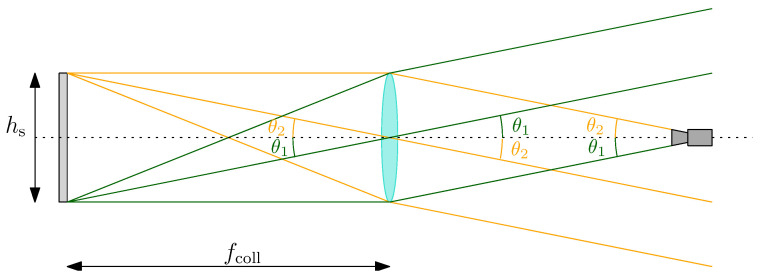
Optical configuration of the components of the TinyV3RSE facility. Note that $\theta_{1}=\theta_{2}=\frac{\theta }{2}$.

**Figure 3 sensors-22-09333-f003:**
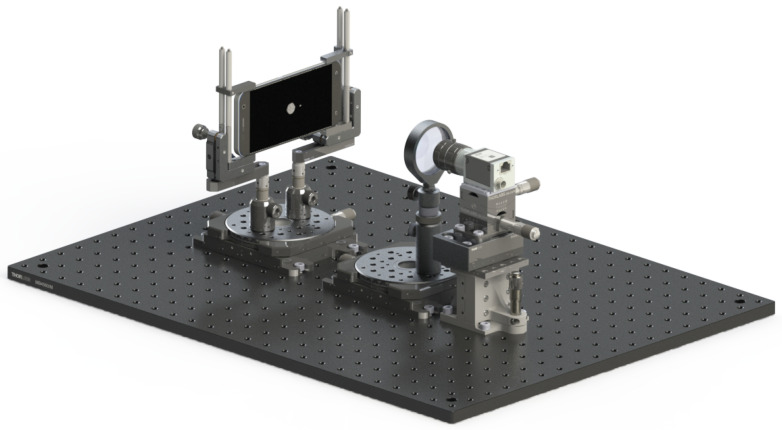
CAD model of TinyV3RSE.

**Figure 4 sensors-22-09333-f004:**
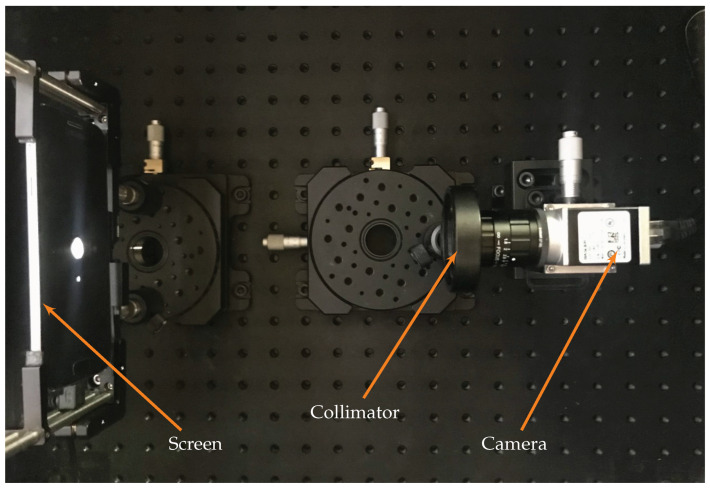
Top view of the TinyV3RSE facility.

**Figure 5 sensors-22-09333-f005:**
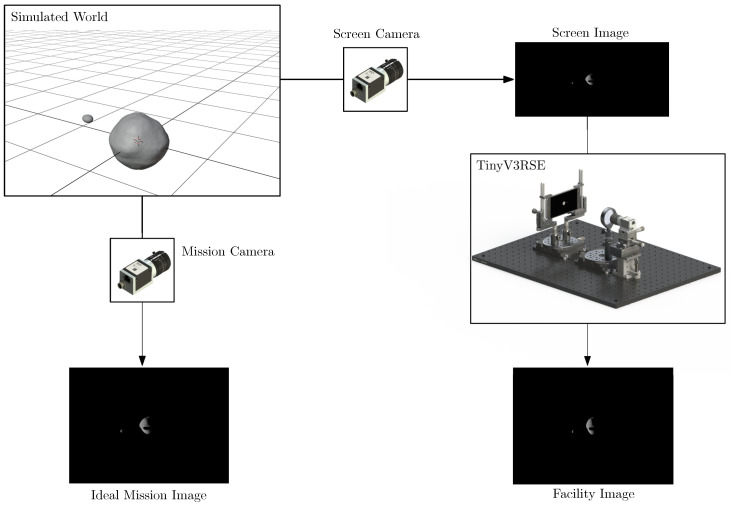
Functional workflow in TinyV3RSE.

**Figure 6 sensors-22-09333-f006:**
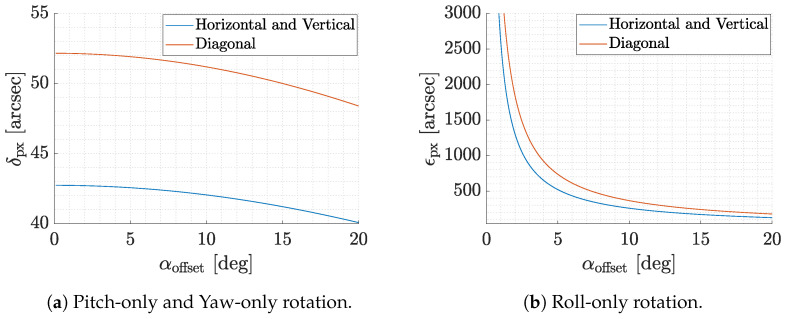
Maximum angular uncertainty for a change in the apparent position of a feature for a displacement in the horizontal, vertical, and diagonal pixel sizes.

**Figure 7 sensors-22-09333-f007:**
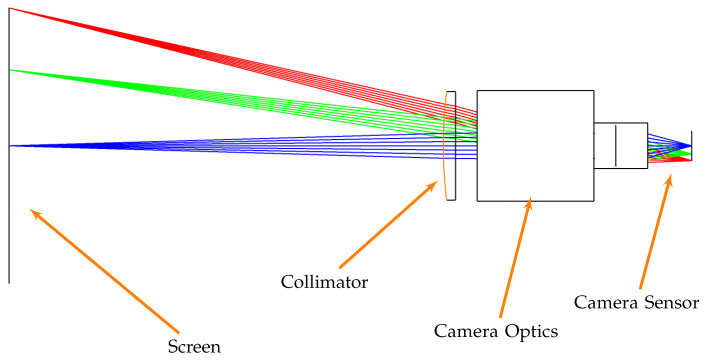
A Zemax^®^ (Tokyo, Japan) model of TinyV3RSE with the camera focused at infinity. Ray colors are associated with the screen-emitting height.

**Figure 8 sensors-22-09333-f008:**
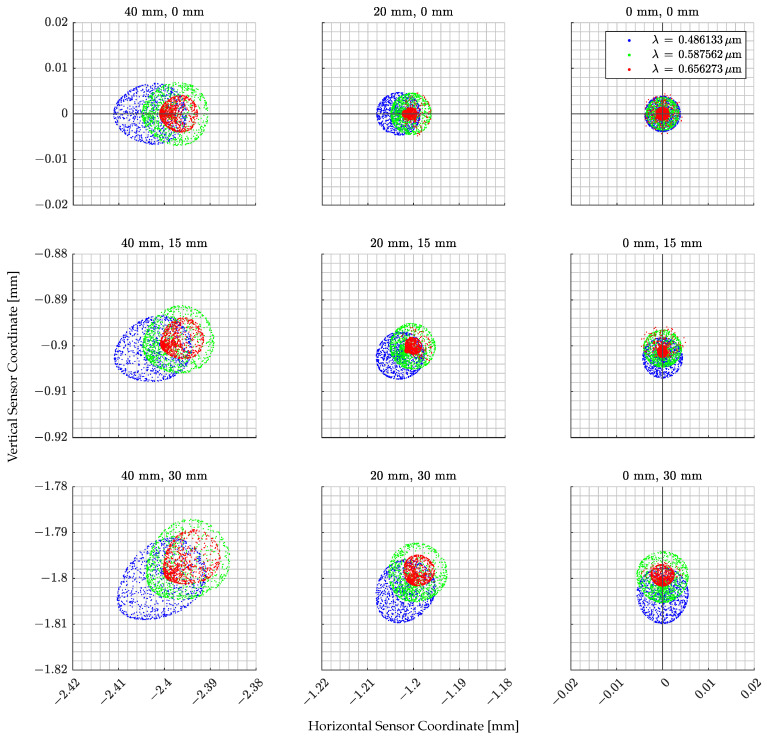
Spot diagram from Zemax^®^. Each plot represents a portion of the camera sensor whose axes ticks are reported on the right and bottom parts of the image for the vertical and horizontal sensor coordinates, respectively. The vertical and horizontal lines show the image axes centered in the camera sensor’s geometrical center. The spot colors are associated with different wavelengths reported in the upper-left legend. The numbers over each plot define where the ray beam is generated on the screen.

**Figure 9 sensors-22-09333-f009:**
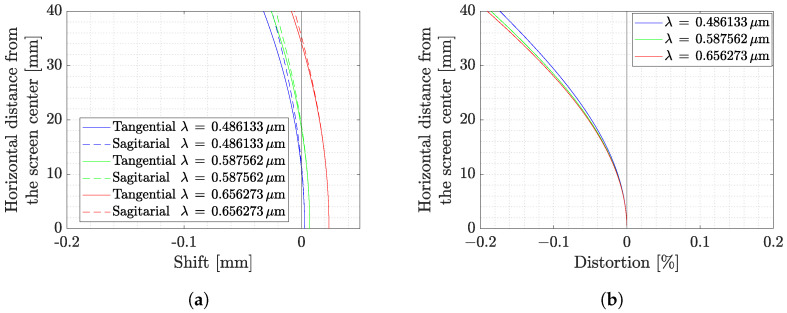
Field curvature and distortion analysis from Zemax^®^ as a function of the horizontal distance. (**a**) Field curvature on tangential and sagittal planes for different wavelengths λ. (**b**) Field distortion for different wavelength λ.

**Figure 10 sensors-22-09333-f010:**
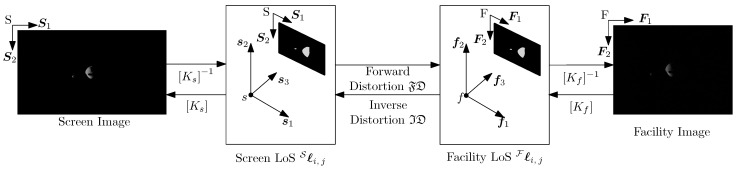
Distortion model.

**Figure 11 sensors-22-09333-f011:**
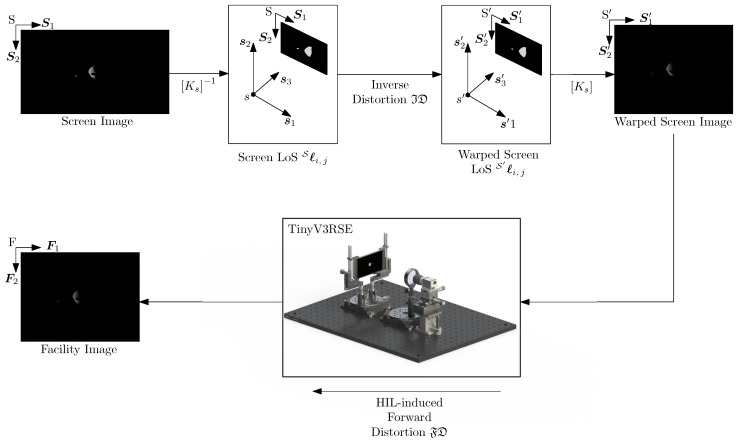
Upstream compensation.

**Figure 12 sensors-22-09333-f012:**
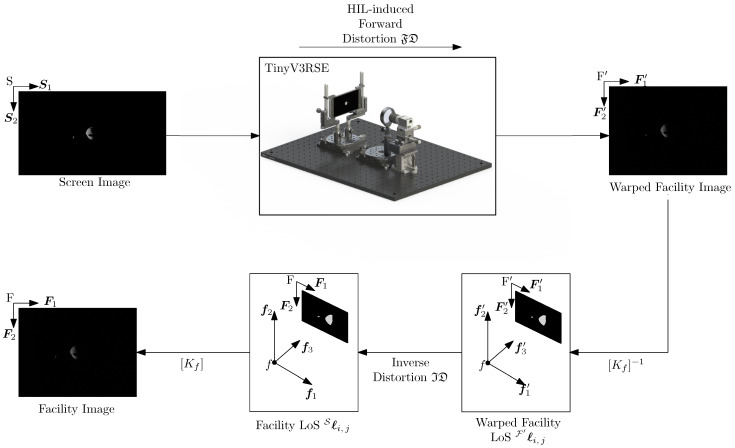
Downstream compensation.

**Figure 13 sensors-22-09333-f013:**
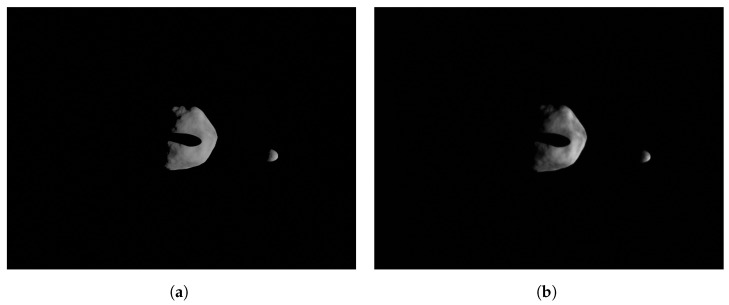
Example of the perfectly rendered image from the mission camera which is used for preliminary design and of the facility image mapped to the mission camera that includes HIL effects. The images show the asteroid binary system Didymos as seen from the Milani CubeSat’s camera (see Piccolo et al. [31] for further details). (**a**) Perfectly rendered image from the mission camera in the virtual world. (**b**) Facility image mapped to the mission camera.

**Figure 14 sensors-22-09333-f014:**
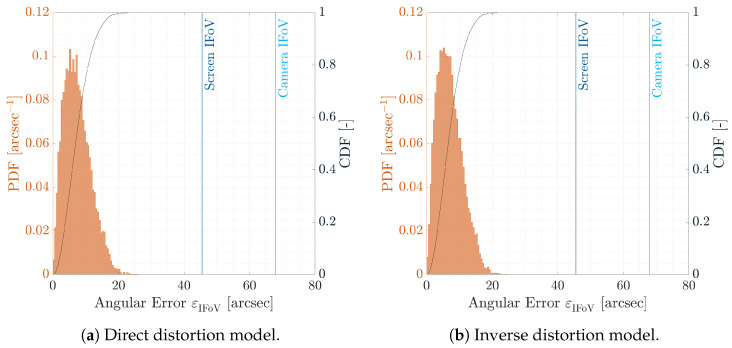
PDF and CDF of the error norm $\varepsilon_{\mathrm{IFoV}}^{}$ for the two distortion models during the calibration. The vertical lines represent the facility iFoV $\theta_{\mathrm{IFoV}}^{f}$, in blue, and the screen iFoV $\theta_{\mathrm{IFoV}}^{s}$, in red.

**Figure 15 sensors-22-09333-f015:**
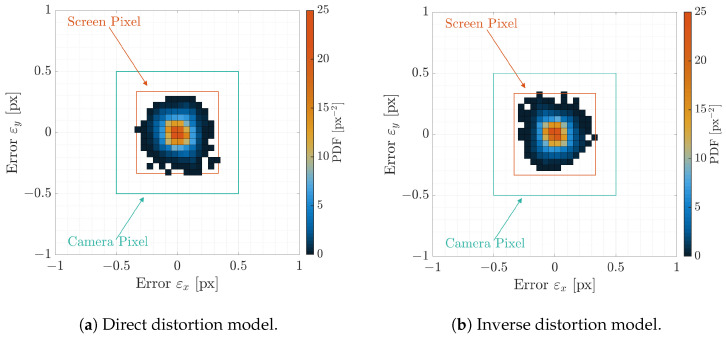
Two-dimensional PDF of the pixel error $\varepsilon ={\left(\varepsilon_{x},\;\varepsilon_{y}\right)}^{T}$ for the two distortion models during calibration. The vertical lines are the facility iFoV $\theta_{\mathrm{IFoV}}^{f}$ and the screen iFoV $\theta_{\mathrm{IFoV}}^{s}$.

**Figure 16 sensors-22-09333-f016:**
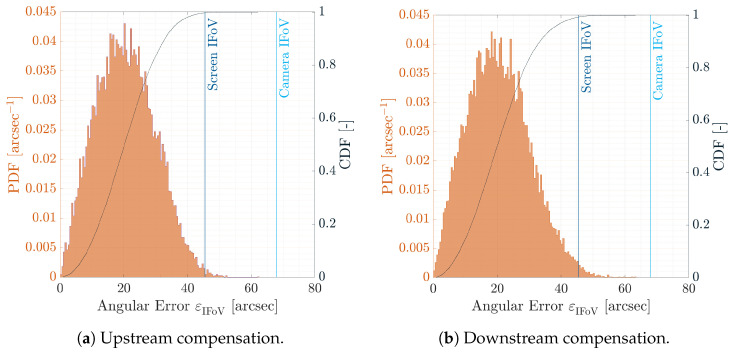
PDF and CDF for the error norm $\varepsilon_{\mathrm{IFoV}}^{}$ during the two proposed compensations. The vertical lines are the facility iFoV $\theta_{\mathrm{IFoV}}^{f}$ and the screen iFoV $\theta_{\mathrm{IFoV}}^{s}$.

**Figure 17 sensors-22-09333-f017:**
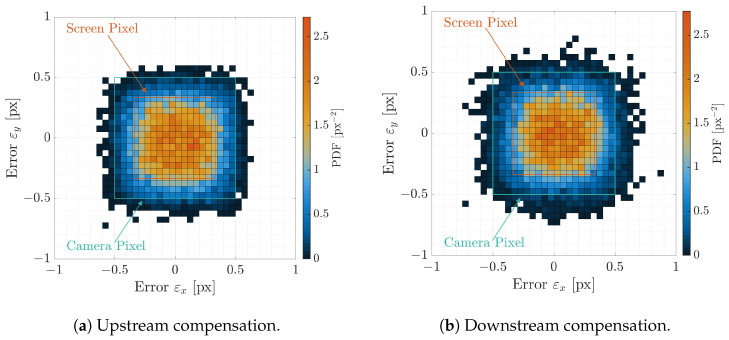
Two-dimensional PDF of the pixel error $\varepsilon ={\left(\varepsilon_{x},\;\varepsilon_{y}\right)}^{T}$ during the two proposed compensations.

**Figure 18 sensors-22-09333-f018:**
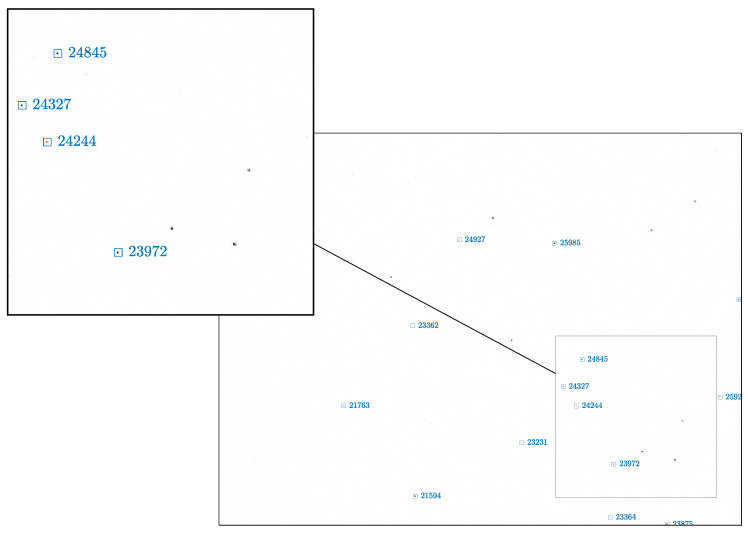
An example image, in reversed colors, of detected stars in a deep-space image for attitude determination. The identified stars are shown in blue boxes with the corresponding identifiers. The image is framed to show the image borders. Details of the image are shown in the box.

**Figure 19 sensors-22-09333-f019:**
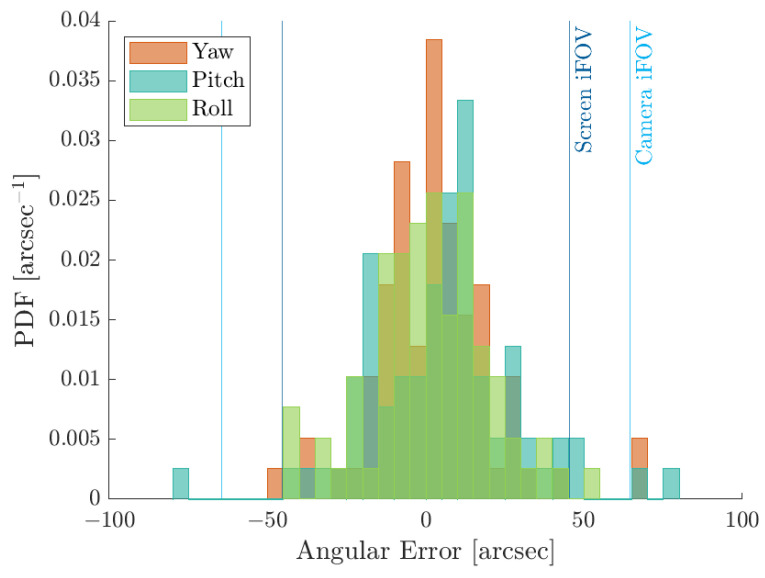
The error in roll, pitch, and yaw between the estimated and true spacecraft-fixed reference frame.

**Figure 20 sensors-22-09333-f020:**
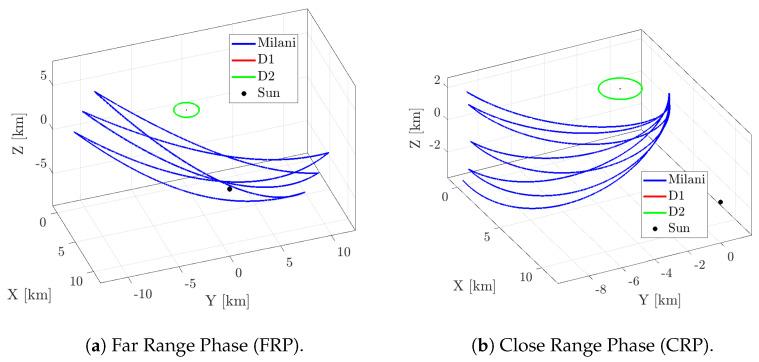
The trajectories for the two Milani’s mission phases (revisited from Pugliatti et al. [41]).

**Figure 21 sensors-22-09333-f021:**
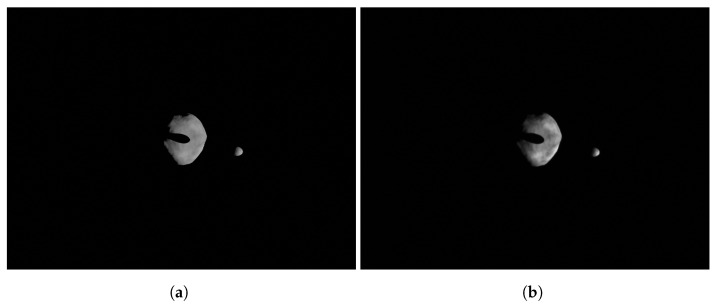
Example of image taken during the FRP. (**a**) Image taken by the camera in the virtual world. (**b**) Image taken by the camera in TinyV3RSE.

**Figure 22 sensors-22-09333-f022:**
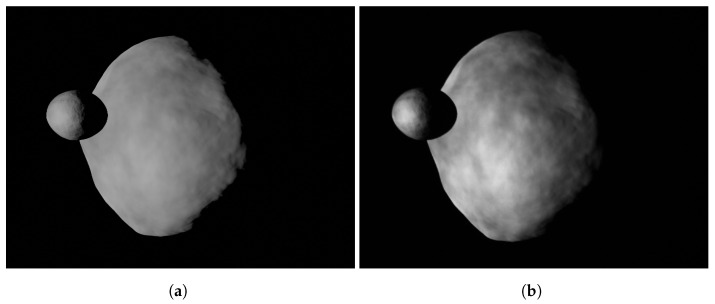
Example image taken during the CRP (see Piccolo et al. [31]). (**a**) Image taken by the camera in the virtual world. (**b**) Image taken by the camera in TinyV3RSE.

**Figure 23 sensors-22-09333-f023:**
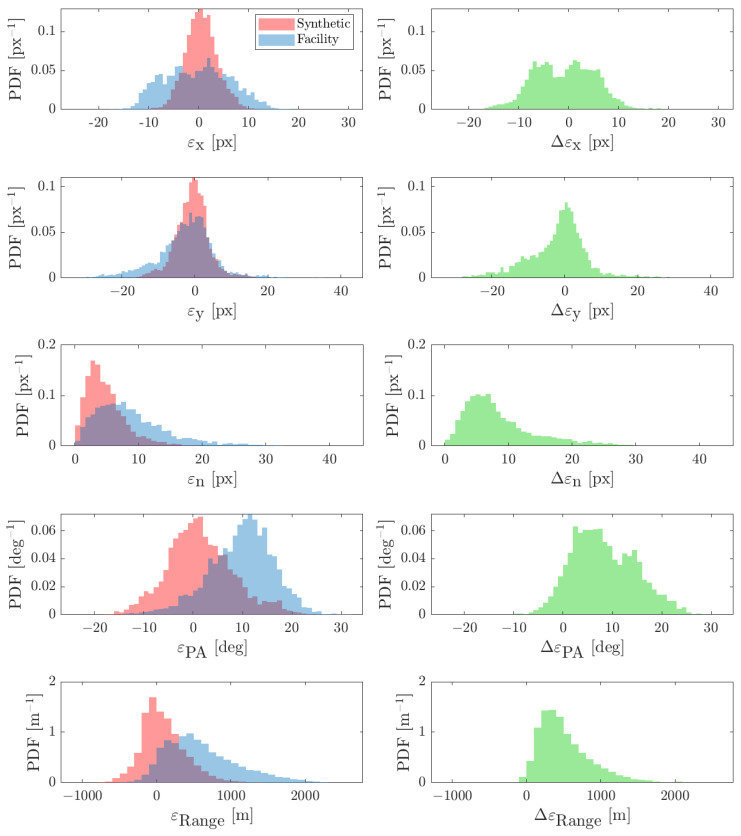
Results for the FRP dataset, shown as PDFs. (**Left**): Estimation error statistics. (**Right**): Difference between the IP output on facility images and the IP output on synthetic images (see Piccolo et al. [31]).

**Figure 24 sensors-22-09333-f024:**
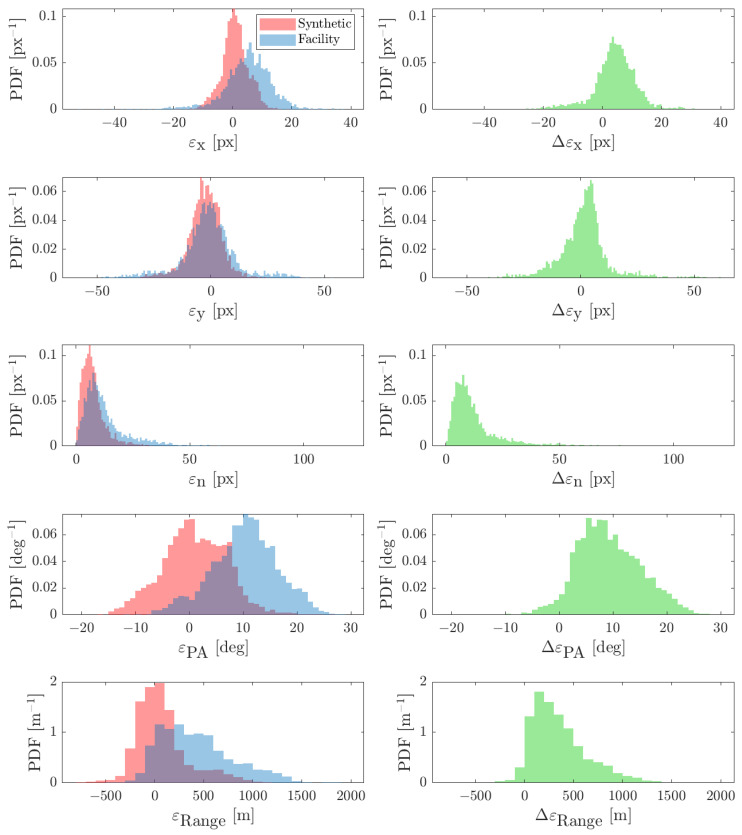
Results for the CRP dataset, shown as PDFs. (**Left**): Estimation error statistics. (**Right**): Difference between the IP output on facility images and the IP output on synthetic images (see Piccolo et al. [31]).

**Table 1 sensors-22-09333-t001:** Mean and covariance of the pixel error $\varepsilon ={\left(\varepsilon_{x},\;\varepsilon_{y}\right)}^{T}$ for the two proposed distortion models.

	Direct Distortion	Inverse Distortion
	Model	Model
$\mu_{\varepsilon }\;\left[\mathrm{px}\right]$	${\left(-4.81\times 10^{-4},\;-5.71\times 10^{-4}\right)}^{T}$	${\left(5.12\times 10^{-4},\;5.05\times 10^{-4}\right)}^{T}$
${\left[P\right]}_{\varepsilon \;\varepsilon }\;\left[{\mathrm{px}}^{2}\right]$	$\left[\begin{array}{cc} 7.23\times 10^{-3} & -6.08\times 10^{-4}\\ -6.08\times 10^{-4} & 7.46\times 10^{-3} \end{array}\right]$	$\left[\begin{array}{cc} 6.51\times 10^{-3} & -5.41\times 10^{-4}\\ -5.41\times 10^{-4} & 8.26\times 10^{-3} \end{array}\right]$

**Table 2 sensors-22-09333-t002:** Mean and covariance of the pixel error $\varepsilon ={\left(\varepsilon_{x},\;\varepsilon_{y}\right)}^{T}$ for the two proposed compensations.

	Upstream	Downstream	Uniform Bivariate
	Compensation	Compensation	Distribution
$\mu_{\varepsilon }\;\left[\mathrm{px}\right]$	${\left(0.018,\;-0.035\right)}^{T}$	${\left(0.039,\;-0.015\right)}^{T}$	${\left(0,\;0\right)}^{T}$
${\left[P\right]}_{\varepsilon \;\varepsilon }\;\left[{\mathrm{px}}^{2}\right]$	$\left[\begin{array}{cc} 0.0542 & -2.9\;10^{-4}\\ -2.9\;10^{-4} & 0.05419 \end{array}\right]$	$\left[\begin{array}{cc} 0.0549 & 7.5\;10^{-4}\\ 7.5\;10^{-4} & 0.0543 \end{array}\right]$	$\left[\begin{array}{cc} 0.05577 & 0\\ 0 & 0.05577 \end{array}\right]$

## Data Availability

Data are available on demand.

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
