# Peer review of "The TinyV3RSE Hardware-in-the-Loop Vision-Based Navigation Facility"

_sensors, 2022, doi:10.3390/s22239333_

Round 1

Reviewer 1 Report

Some major modifications is listed as follows.

 1)      FIG. 14 in Chapter 6 shows the general interval of the error distribution. The error satisfies the normal distribution, but the CDF result is not explained in detail. Please explain the role of CDF in this figure and why the CDF diagram is drawn.

2)      The ordinate in Figure 23 does not show units, please fill in the corresponding units.

3)      It can be seen that the error is significantly reduced after compensation, but why does the amount of data displayed in the compensated PDF chart increase? Does it refer to the same experimental result? Please provide additional explanations.

4)      Shorten the manuscript as much as you could to make it more concise.

5)      Could you analyse the computational complexity of the vision-based navigation algorithm to show its feasibility?

Author Response

Review document attached

Reviewer 2 Report

The paper “The Tiny V3RSE Hardware-In-The-Loop Vision-Based Navigation Facility" is devoted to description of the laboratory facility and to study of its performance. The paper can be of interest for GNC-specialists. The presented work is of high quality, two application examples are demonstrated. The text is quite well-written and clear, but it should be improved in some parts before the acceptance for publication.

 First of all, it seems that the length of the paper can be reduced to increase the readability. It would be better if the authors will shorten the Introduction and the overview in the Section 3. Also, some well-known calibration stages coould be explained briefly. The authors should avoid repetitive statements in the text (for example, that geometrical distortions are not present in orbit) and condense the descriptions (for example, descriptions of upstream and downstream compensation procedures).

 Page 9, first paragraph. Second sentence is not completed.

Page 9. Last sentence is inconsistent, it should be corrected.

Do the authors observe the Fraunhofer diffraction effect when the ratio between the pixels of the sensor and the screen is 1:1.39, as on a diffraction grating? Can this effect be evaluated analytically?

In eq. 30, it is not clear what the last term “mu_h*0_3x1” means. Is it just a zero vector? Why then is the point vector “p” a function of “mu”? In eq. (31), in the right part, this zero vector is just disappeared. Can the authors add comments on it? The text after eq. (31) should be rewritten – too many points and the third is twofold, it is hard to follow the idea. The authors should condense their statements or provide some summary.

In Fig. 15 the two-dimensional PDF is presented, but the bar label is “Probability”, not PDF. Moreover, it seems that the probability values are too high (the highest is 25%?) and it is not normalized (the sum is not 100%). The same comment is for Fig. 17. Can the authors add some comments?

The Y-axis for Fig. 19, 23 and 24 is without label.

Author Response

Review document attached.
